# SPIN1 promotes tumorigenesis by blocking the uL18 (universal large ribosomal subunit protein 18)-MDM2-p53 pathway in human cancer

Ziling Fang[1†], Bo Cao[1], Jun-Ming Liao[1,2], Jun Deng[1†], Kevin D Plummer[1‡], Peng Liao[1], Tao Liu[1§], Wensheng Zhang[3], Kun Zhang[3], Li Li[4], David Margolin[5], Shelya X Zeng[1]*, Jianping Xiong[6]*, Hua Lu[1]*

[1]Department of Biochemistry and Molecular Biology, Tulane University School of Medicine, New Orleans, United States; [2]School of Dentistry at Case Western University, Cleveland, United States; [3]Department of Computer Science, Bioinformatics Facility of Xavier RCMI Center of Cancer Research, Xavier University of Louisiana, New Orleans, United States; [4]Laboratory of Translational Cancer Research, Ochsner Clinical Foundation, New Orleans, United States; [5]Department of Colon and Rectal Surgery, Ochsner Clinical Foundation, New Orleans, United States; [6]Department of Oncology, The First Affiliated Hospital of Nanchang University, Nanchang, China

*For correspondence:
szeng@tulane.edu (SXZ);
jpxiong@ncu.edu.cn (JX);
hlu2@tulane.edu (HL)

Present address: †Department of Oncology, The First Affiliated Hospital of Nanchang University, Nanchang, China; ‡Southern New Hampshire University, New Hampshire, Unites States; §Department of Pediatrics, The First Affiliated Hospital of Nanchang University, Nanchang, China

Competing interests: The authors declare that no competing interests exist.

**Abstract** Ribosomal proteins (RPs) play important roles in modulating the MDM2-p53 pathway. However, less is known about the upstream regulators of the RPs. Here, we identify SPIN1 (Spindlin 1) as a novel binding partner of human RPL5/uL18 that is important for this pathway. SPIN1 ablation activates p53, suppresses cell growth, reduces clonogenic ability, and induces apoptosis of human cancer cells. Mechanistically, SPIN1 sequesters uL18 in the nucleolus, preventing it from interacting with MDM2, and thereby alleviating uL18-mediated inhibition of MDM2 ubiquitin ligase activity toward p53. SPIN1 deficiency increases ribosome-free uL18 and uL5 (human RPL11), which are required for SPIN1 depletion-induced p53 activation. Analysis of cancer genomic databases suggests that SPIN1 is highly expressed in several human cancers, and its overexpression is positively correlated with poor prognosis in cancer patients. Altogether, our findings reveal that the oncogenic property of SPIN1 may be attributed to its negative regulation of uL18, leading to p53 inactivation.

DOI: https://doi.org/10.7554/eLife.31275.001

## Introduction

The well-documented tumor suppressor p53, referred as 'the guardian of the genome', is activated upon exposure to a myriad of cellular stresses. While loss of wild-type p53 causes fatal damages to the genome, it is not surprising that the *TP53* gene is mutated in more than 50% human cancers, and the functions of p53 are often impeded through various mechanisms in the remainder (*Levine and Oren, 2009*). One predominant negative regulator of p53 is the E3 ubiquitin ligase MDM2, an oncoprotein that conceals the N-terminal transcriptional activation (TA) domain of p53 (*Oliner et al., 1993*) and deactivates this protein by either abrogating its transcriptional activity, or inducing its nuclear export and ubiquitination (*Oliner et al., 1993*; *Haupt et al., 1997*; *Kubbutat et al., 1997*; *Fuchs et al., 1998*). A plethora of cellular stress could stabilize p53 by

blocking the MDM2-p53 feedback loop (*Kim et al., 2014*). For example, p19[ARF] inhibits MDM2-mediated p53 ubiquitination and proteolysis by associating with MDM2 (*Zhang et al., 1998*).

Another pathway is the so-called ribosomal proteins (RPs)-MDM2-p53 pathway (*Zhang and Lu, 2009*; *Warner and McIntosh, 2009*). Accumulating evidence has continuingly verified this pathway as an emerging mechanism for boosting p53 activation in response to ribosomal stress or nucleolar stress over the past decade (*Sun et al., 2007*; *Sun et al., 2008*; *Dai et al., 2004*; *He et al., 2016*; *Bai et al., 2014*). Ribosomal stress is often triggered by aberrant ribosome biogenesis caused by nutrient deprivation, inhibition of rRNA synthesis, or malfunction of ribosomal proteins and/or nucleolar proteins (*Zhang and Lu, 2009*; *Warner and McIntosh, 2009*; *Sun et al., 2007*; *Sun et al., 2008*; *Fumagalli et al., 2009*; *Bhat et al., 2004*). Earlier studies showed that disruption of ribosomal biogenesis induces translocation of a series of ribosomal proteins, including uL18 (human RPL5), uL5 (human RPL11), uL14 (human RPL23), eS7 (human S7) and uS11 (human S14) (*Ban et al., 2014*), from the nucleolus to the nucleoplasm and bind to MDM2, blocking its ability to ubiquitinate p53 and consequently stabilizing p53 to maintain cellular homeostasis (*Dai et al., 2004*; *Lohrum et al., 2003*; *Dai and Lu, 2004*; *Zhou et al., 2013*; *Chen et al., 2007*; *Zhang et al., 2003*; *Jin et al., 2004*). Although there are a few proteins that have been identified to regulate this RPs-MDM2-p53 pathway, such as PICT-1 inhibition of uL5 (*Sasaki et al., 2011*; *Uchi et al., 2013*) and SRSF1 activation of uL18 (*Fregoso et al., 2013*), it still remains to be determined if there are more proteins that can regulate the RPs-MDM2-p53 pathway. In this present study, we identified SPIN1 as a new uL18 inhibitory regulator.

SPIN1, a new member of the SPIN/SSTY family, was originally identified as a highly expressed protein in ovarian cancer (*Yue et al., 2004*). The oncogenic potential of SPIN1 was later supported by the observation that overexpression of SPIN1 increases transformation and tumor growth ability of NIH3T3 cells (*Gao et al., 2005*). Signaling pathways responsible for SPIN1 functions include PI3K/Akt, Wnt and RET that are highly relevant to tumorigenesis (*Chen et al., 2016*; *Wang et al., 2012*; *Franz et al., 2015*). In addition, SPIN1 acts as a reader of histone H3K4me3 and stimulates the transcription of ribosomal RNA-encoding genes (*Bae et al., 2017*; *Su et al., 2014*; *Wang et al., 2011*), suggesting its role in rRNA synthesis.

In screening uL18-associated protein complexes using co-immunoprecipitation followed by mass spectrometry, we identified SPIN1 as one of the potential uL18 binding proteins. We confirmed the specific interaction of SPIN1 with uL18, but not with uL5 or uL14, and found out that by binding to uL18, SPIN1 prevents the inhibition of MDM2 by uL18 and promotes MDM2-mediated p53 ubiquitination and degradation. Also, SPIN1 knockdown induced ribosomal stress by facilitating the release of ribosome-free uL18 or uL5, accompanying p53 activation. Furthermore, SPIN1 knockdown inhibited cell proliferation and induced apoptosis in a predominantly p53-dependent manner in vitro and in vivo, consequently suppressing tumor growth in a xenograft model. Therefore, these results for the first time demonstrate that SPIN1 can regulate the RP-MDM2-p53 pathway by directly interacting with uL18, and suggest SPIN1 as a potential molecule target in this pathway for developing anti-cancer therapy in the future.

## Results

### SPIN1 interacts with uL18

Our and others' studies previously demonstrated that uL18 can stabilize p53 by binding to MDM2 and inhibiting its E3 ligase activity toward p53 (*Dai and Lu, 2004*; *Bursać et al., 2012*). In order to identify potential upstream regulators that may modulate the uL18-MDM2-p53 circuit, we performed co-immunoprecipitation (co-IP) using HEK293 cells that stably expressed Flag-uL18 with the anti-Flag antibody, and the co-immunoprecipitated proteins were cut out for mass spectrometric (MS) analysis (*Figure 1A*). The MS results not only revealed several previously described p53 regulatory proteins, such as MYBBP1A, PRMT5 and SRSF1, as uL18 binding proteins (*Table 1*), but also identified SPIN1 as a novel uL18-binding protein candidate that was previously shown to play a role in tumorigenesis and rDNA transcription (*Wang et al., 2012*, *2011*).

Next, we confirmed the interaction between SPIN1 and uL18 by performing a series of reciprocal co-IP assays. As expected, ectopic SPIN1 was specifically pulled down by ectopic uL18 and vice versa in HCT116[p53-/-] cells (*Figure 1B and C*). Their interaction was also verified in HEK293 cells

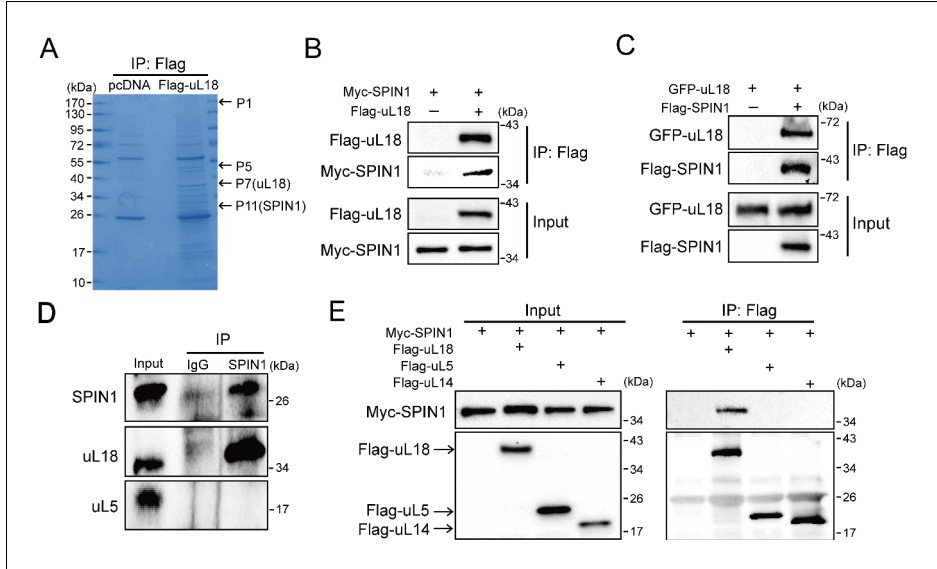

**Figure 1.** SPIN1 binds to uL18, but not uL5, or uL14. (**A**) Identification of SPIN1 as a candidate of uL18 binding protein by immunopurification and mass spectrometric analysis. Lysates from HEK 293 cells were immunoprecipitated with the anti-Flag antibody. Bound proteins were visualized on a coomassie staining SDS-PAGE gel. Several bands were excised and subjected to mass spectrometry. One of them was identified as SPIN1 (Spindlin 1). The polypeptides identified from these bands are listed in *Table 1*. (**B**) and (**C**) SPIN1 interacts with uL18. (**B**) HCT116^p53-/- cells were transfected with plasmids encoding Myc-SPIN1 and Flag-uL18, and 48 hr later cell lysates were collected for immunoprecipitation (IP) analysis using the anti-Flag antibody. (**C**) HCT116^p53-/- cells were transfected with plasmids encoding Flag-SPIN1 and GFP-uL18 for 48 hr and harvested for IP/WB analysis with indicated antibodies. (**D**) The interaction between endogenous SPIN1 and uL18. The HEK 293 cell lysates were immunoprecipitated with anti-SPIN1 or control immunoglobulin G (IgG), followed by WB analysis with anti-SPIN1, anti-uL18 and anti-uL5. (**E**) SPIN1 was specifically co-immunoprecipitated by uL18, but not uL5 or uL14. H1299 cells were co-transfected with Myc-SPIN1 and Flag-uL18, Flag-uL5 or Flag-uL14 as indicated and subjected to IP with the anti-Flag antibody, followed by WB analysis with indicated antibodies.
DOI: https://doi.org/10.7554/eLife.31275.002
The following figure supplement is available for figure 1:

**Figure supplement 1.** SPIN1 interacts with uL18 in HEK293 cells.
DOI: https://doi.org/10.7554/eLife.31275.003

---

(*Figure 1—figure supplement 1*). Also, we validated the interaction between endogenous SPIN1 and uL18 in HEK293 cells using anti-SPIN1 antibody (*Figure 1D*). Interestingly, only uL18, but not uL5, was co-immunoprecipitated with SPIN1. In line with this result, when comparing ectopic Flag-uL18 with Flag-uL5 and Flag-uL14, we found that only uL18, but not the other RPs, could pull down Myc-SPIN1 (*Figure 1E*), further bolstering the specific interaction between uL18 and SPIN1. Taken together, these results demonstrate that SPIN1 specifically binds to uL18, but not uL5 or uL14, in cancer cells.

**Table 1.** uL18-associated polypeptides identified from mass spectrometry analysis of proteins as shown in *Figure 1A*.

| Gel slice number | Protein | Accession | Molecular weight | Score |
|---|---|---|---|---|
| P1 | Myb-binding protein 1A (MYBBP1A) | gi\|6959304 | 149727 | 149 |
| P5 | Protein arginine N-methyltransferase 5 (PRMT5) | gi\|2323410 | 72685 | 121 |
| P7 | 60S ribosomal protein L5 uL18(RPL5) | gi\|14591909 | 34569 | 1014 |
| P11 | Serine/arginine-rich splicing factor 1 (SRSF1) | gi\|5902076 | 27746 | 75 |
| P11 | Spindlin 1 (SPIN1) | gi\|5410330 | 29602 | 95 |

DOI: https://doi.org/10.7554/eLife.31275.004

## SPIN1 knockdown inhibits proliferation and induces apoptosis of cancer cells by activating p53

Previous and recent studies showed that SPIN1 is a potential oncogene (*Chen et al., 2016*; *Wang et al., 2012*; *Chen et al., 2017*), and uL18 can stabilize p53 by binding to MDM2 (*Dai and Lu, 2004*). We therefore wondered if the interaction between SPIN1 and uL18 could confer any role to SPIN1 in regulation of the p53 pathway. First, we determined if depletion of SPIN1 might affect p53-dependent cellular outcomes. Interestingly, we found that knockdown of SPIN1 dramatically elevates p53 protein level in several wild-type p53-containing cells, including U2OS, H460 and HCT116$^{P53+/+}$ cells (*Figure 2A*), without affecting *TP53* mRNA expression (*Figure 2B*). Consistently, protein and mRNA levels of p53 target genes, such as p21 and PUMA, were also increased in response to SPIN1 knockdown (*Figure 2A and B*). Moreover, the effects of SPIN1 siRNA on p53 activity were dose-dependent (*Figure 2—figure supplement 1A and B*). The effect was unlikely due to off-target effects of siRNA, as ectopic expression of FLAG-SPIN1 reversed p53 activation by siRNA-mediated knockdown of SPIN1 in HCT116$^{P53+/+}$ (*Figure 2—figure supplement 1C*). Conversely, overexpression of SPIN1 in HCT116$^{P53+/+}$ decreased the protein levels of p53 and its targets, such as p21 and PUMA, and the mRNA levels of these target genes, without affecting *TP53* mRNA level (*Figure 2C and D*).

We next generated both HCT116$^{P53+/+}$ and HCT116$^{P53-/-}$ cell lines that express scramble shRNA or SPIN1 shRNA to evaluate biological outcomes of SPIN1 knockdown. As illustrated in *Figure 2E*, the expression of p53 and some of its target genes were markedly induced when SPIN1 was knocked down by its specific shRNA in HCT116$^{P53+/+}$cells, but not in HCT116$^{P53-/-}$cells. Using these cell lines for cell viability assays, we observed that SPIN1 ablation more dramatically represses the cell viability of HCT116$^{P53+/+}$ than that of HCT116$^{P53-/-}$ cells (*Figure 2F*). In line with this observation, SPIN1 depletion also led to more predominant reduction of HCT116$^{P53+/+}$ colonies than that of HCT116$^{P53-/-}$ colonies, although both of the reductions were statistically significant (*Figure 2G*). Furthermore, the percentage of cells undergoing apoptosis caused by SPIN1 shRNAs was much higher in HCT116$^{P53+/+}$ cells than in HCT116$^{P53-/-}$ cells, as measured by sub-G1 population (*Figure 2H*). Consistently, induction of apoptosis by SPIN1 knockdown was also evidenced in Annexin V assay in U2OS cells (*Figure 2I*). Collectively, these data suggest that SPIN1 plays an oncogenic role at least partially by inactivating the p53 pathway, although SPIN1 may also possess a p53-independent role in cancer cell growth and survival.

## SPIN1 promotes p53 degradation by enhancing MDM2-mediated ubiquitination

Since SPIN1 knockdown affected only the protein, but not the mRNA, levels of p53 (*Figure 2A–2D*), we next sought to determine the underlying mechanism. We first performed a cycloheximide-chase experiment using HCT116$^{P53+/+}$ cells. As shown in *Figure 3A and B*, knockdown of SPIN1 markedly prolonged p53's half-life from 35 mins to 56 mins, as compared to scramble siRNA. Inversely, ectopic SPIN1 greatly shortened p53's half-life, from 39 mins to ~22 mins (*Figure 3C and D*). To further evaluate the effect of SPIN1 on MDM2-mediated p53 ubiquitination, which is the main mechanism responsible for p53 turnover (*Dai et al., 2004*; *Dai and Lu, 2004*; *Zhang et al., 2003*; *Dai et al., 2006*), we then performed an in vivo ubiquitination assay by transfecting HCT116$^{P53-/-}$ cells with plasmids indicated in *Figure 3E*. The results clearly showed that ectopic SPIN1 enhances MDM2-mediated p53 ubiquitination in a dose-dependent manner. Consistently, co-transfection of SPIN1 with MDM2 led to a stronger reduction of p53 protein levels, which was abrogated by proteasome inhibitor MG132 (*Figure 3F*). Interestingly, the induction of p53 degradation by SPIN1 was MDM2-dependent, as overexpression of SPIN1 failed to repress ectopic p53 protein expression in *Trp53* and *Mdm2* double knockout MEF cells (*Figure 3G*). Together, these results demonstrate that SPIN1 reduces p53 stability by enhancing MDM2-mediated ubiquitination and degradation.

## SPIN1 prevents uL18 from MDM2 binding by sequestering it in the nucleolus

Besides its role as a component of ribosome, uL18 has some well-established extra-ribosomal functions, acting as a bridge in connecting p53 activation to cellular stress response machinery (*Zhang and Lu, 2009*; *Warner and McIntosh, 2009*). Upon ribosomal stress, uL18 can translocate

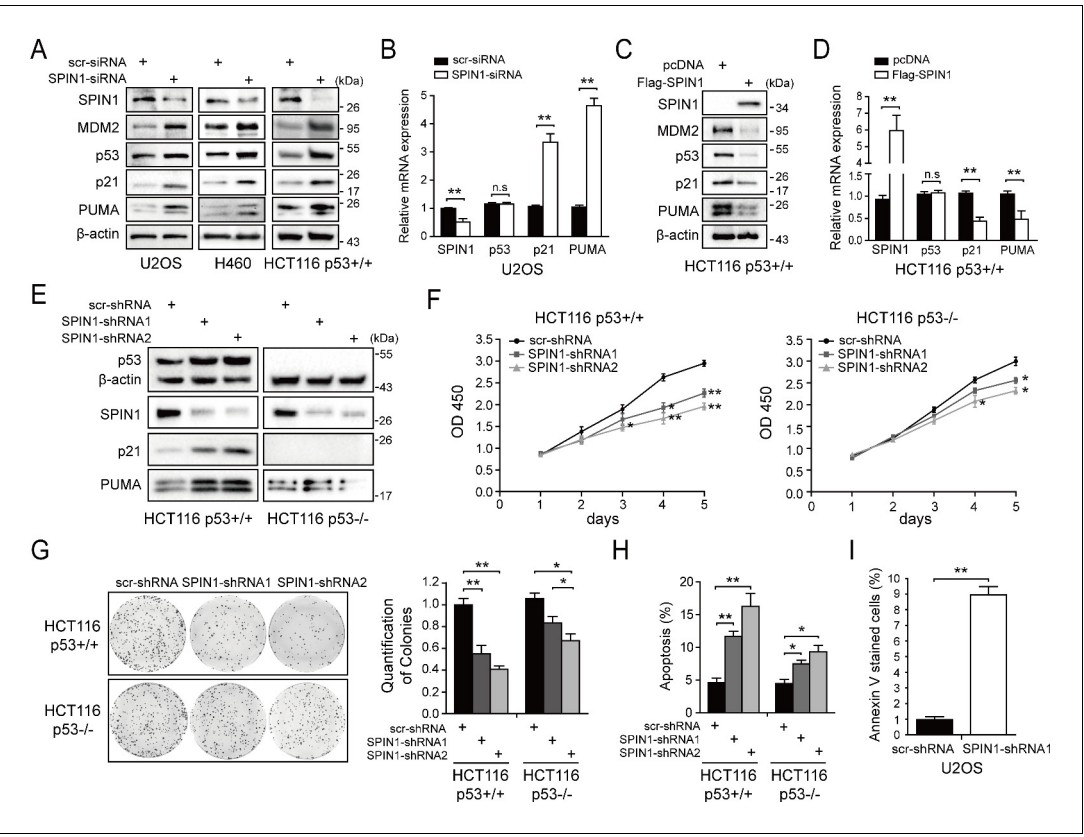

**Figure 2.** SPIN1 knockdown inhibits cell proliferation and induces apoptosis. (**A**) SPIN1 knockdown induces protein levels of p53 and its target genes. U2OS, H460 and HCT116$^{p53+/+}$ cells were transfected with scramble siRNA (scr-siRNA) or SPIN1 siRNA and harvested 48 hr post-transfection for WB analysis with indicated antibodies. (**B**) SPIN1 knockdown induces mRNA levels of p53 target genes without effect on *TP53* mRNA level. U2OS cells were transfected with scramble siRNA (scr-siRNA) or SPIN1 siRNA, and harvested 72 hr post-transfection for RT-qPCR (mean ± SEM, n = 2). (**C**) SPIN1 overexpression reduces protein levels of p53 and its target genes. HCT116$^{p53+/+}$ cells were transfected with pcDNA or Flag-SPIN1 and harvested 48 hr post-transfection for WB analysis with indicated antibodies. (**D**) SPIN1 overexpression reduces mRNA levels of p53 target genes without effect on *TP53* mRNA levels. HCT116$^{p53+/+}$ cells were transfected with pcDNA or Flag-SPIN1 and harvested 72 hr post-transfection for RT-qPCR (mean ± SEM, n = 2). (**E**) Knockdown of SPIN1 causes p53-dependent induction of p21 and PUMA. The protein levels of p53 and its targets in HCT116$^{p53+/+}$ cells and HCT116$^{p53-/-}$ cells that stably express scramble shRNA (scr-shRNA) or SPIN1 shRNAs were detected by WB analysis with indicated antibodies. (**F**) SPIN1 knockdown suppresses cell survival. HCT116$^{p53+/+}$ and HCT116$^{p53-/-}$ cells that stably expressed scramble or SPIN1 shRNAs were seeded in 96-well plate and cell viability was evaluated every 24 hr by CCK-8 assays (mean ± SEM, n = 2). (**G**) Knockdown of SPIN1 inhibits clonogenic ability of colorectal cancer cells, more significantly when the cells harbor wild-type p53. HCT116$^{p53+/+}$ cells and HCT116$^{p53-/-}$cells that stably expressed scramble or SPIN1 shRNAs were seeded on 60 mm plates. Puromycin selection was performed for 14 days. Colonies were fixed with methanol, and visualized by staining with crystal violet (mean ± SEM, n = 3). (**H**) The effect of SPIN1 knockdown on apoptosis of HCT116$^{p53+/+}$ cells and HCT116$^{p53-/-}$cells that stably expressed scramble or SPIN1 shRNAs (mean ± SEM, n = 3). (**I**) U2OS cells were transfected with scramble or SPIN1 shRNA and incubated in IncuCyte S3 chamber in the presence of IncuCyte Annexin V Green Reagent for apoptosis. Positively stained cells were determined using IncuCyte analysis software. *p<0.05, **p<0.01 by two-tailed *t*-test (**C, D, G, H,I**).

DOI: https://doi.org/10.7554/eLife.31275.005

The following figure supplement is available for figure 2:

**Figure supplement 1.** SPIN1 knockdown increase p53 and its targets protein levels in a dose-dependent manner.
DOI: https://doi.org/10.7554/eLife.31275.006

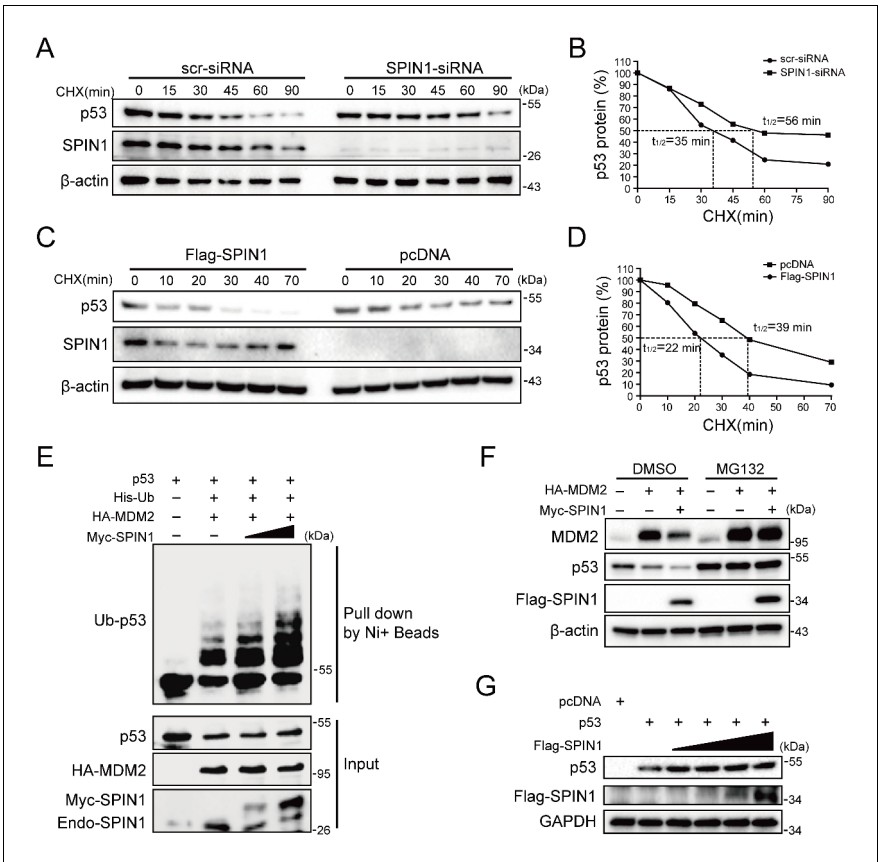

**Figure 3.** SPIN1 reduces p53 stability by enhancing MDM2-mediated ubiquitination. (**A**) and (**B**) p53-half-life is increased by SPIN1 knockdown. (**A**) HCT116[p53+/+] cells transfected with scramble or SPIN1 siRNA for 48 hr, were treated with 100 µg/ml of cycloheximide (CHX), and harvested at different time points as indicated. The p53 protein was detected by WB analysis, quantified by densitometry and plotted against time to determine p53-half-lives (**B**). (**C**) and (**D**) SPIN1 overexpression shortens the half-life of p53. HCT116[p53+/+] cells transfected with pcDNA or Flag-SPIN1 for 48 hr were treated with 100 µg/ml of cycloheximide and harvested at indicated time points for WB analysis with indicated antibodies (**C**). The intensity of each band was quantified, and normalized with β-actin and plotted (**D**). (**E**) SPIN1 promotes MDM2-induced p53 ubiquitination. HCT116[p53-/-] cells were transfected with combinations of plasmids encoding HA-MDM2, p53, His-Ub or Myc-SPIN1, and treated with MG132 for 6 hr before being harvested for in vivo ubiquitination assay. Bound and input proteins were detected by WB analysis with indicated antibodies. (**F**) SPIN1 enhances MDM2-mediated p53 proteasomal degradation. HCT116[p53+/+] cells were transfected with plasmids encoding HA-MDM2 and Flag-SPIN1, and treated with MG132 for 6 hr before harvested, followed by WB analysis with antibodies as indicated. (**G**) Ectopic SPIN1 does not change p53 protein level without MDM2. MEF[p53-/-; Mdm2-/-] cells were transfected with combinations of plasmids encoding p53 with or without Flag-SPIN1, followed by WB analysis using antibodies as indicated.
DOI: https://doi.org/10.7554/eLife.31275.007

from the nucleolus to the nucleoplasm of a cell, where it binds to MDM2 (*Dai and Lu, 2004*; *Zhou et al., 2012*), leading to stabilization of p53 and consequently p53-dependent cell growth arrest, apoptosis or senescence. We then investigated if SPIN1 might regulate this function of uL18, since SPIN1 could bind to uL18 (*Figure 1*), knockdown of SPIN1 led to p53 activation (*Figure 2*), and SPIN1 stimulated MDM2-mediated p53 ubiquitination (*Figure 3*). First, as expected (*Dai and Lu, 2004*), overexpression of uL18 induced the protein levels of p53 and its targets, such as p21 and MDM2, in wild-type p53-containing U2OS cells (*Figure 4A*). This induction of the p53 pathway by uL18 was markedly reduced by co-transfected SPIN1 (*Figure 4A*). Since the effect of uL18 on p53 is through uL18's interaction with MDM2 and consequent inhibition of its E3 ligase activity toward p53 (*Dai and Lu, 2004*), we tested if SPIN1 may affect uL18-MDM2 interaction. Interestingly, our co-immunoprecipitation result showed that ectopic Myc-SPIN1 dramatically reduces the amount of

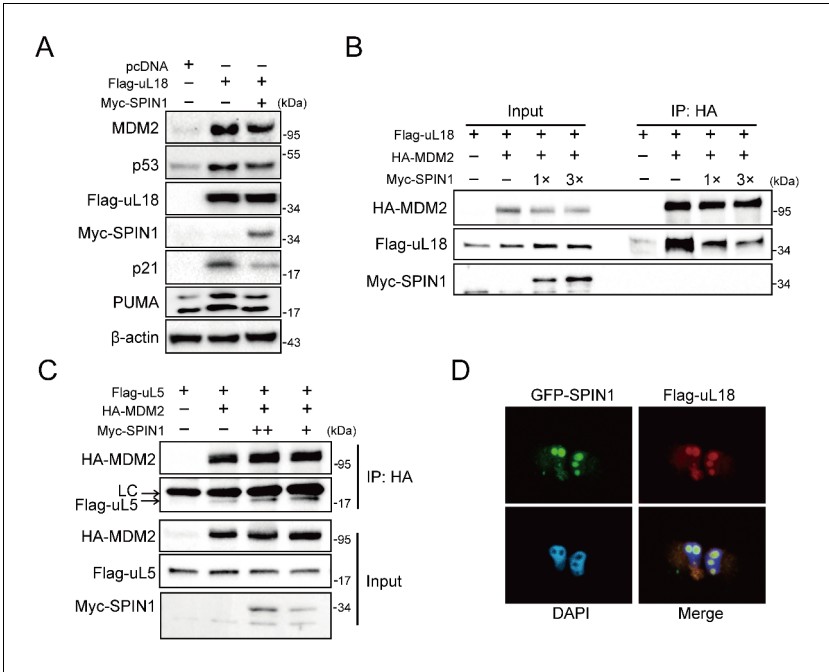

**Figure 4.** SPIN1 blocks uL18-MDM2 interaction by sequestering uL18 in the nucleolus. (**A**) SPIN1 overexpression attenuates p53 activation induced by ectopic uL18. U2OS cells were co-transfected with plasmids encoding Flag-uL18 or Myc-SPIN1 for 36 hr and harvested for WB analysis with indicated antibodies. (**B**) Overexpression of SPIN1 disrupts the uL18-MDM2 binding. Lysates were prepared from HCT116$^{p53-/-}$ cells co-transfected with HA-MDM2, Flag-uL18, Myc-SPIN1 or the corresponding empty vectors for 48 hr and analyzed by immunoprecipitated with the anti-HA antibody. Immunoprecipitates and 5% of inputs were immunoblotted with the indicated antibodies. (**C**) Overexpression of SPIN1 fails to disrupt the uL5-MDM2 interaction. Lysates were prepared from HCT116$^{p53-/-}$ cells co-transfected with HA-MDM2, Flag-uL5 and Myc-SPIN1 for 48 hr and analyzed by immunoprecipitated with the anti-HA antibody. Immunoprecipitates and 5% of inputs were immunoblotted with the indicated antibodies. (LC: light chain). (**D**) SPIN1 and uL18 co-localize in the nucleolus. H1299 cells were transfected with GFP-SPIN1 and Flag-uL18 for 36 hr and then immunostained with the anti-Flag antibody (red), and counterstained with DAPI.
DOI: https://doi.org/10.7554/eLife.31275.008
The following figure supplement is available for figure 4:

**Figure supplement 1.** SPIN1 does not bind to MDM2, and SPIN1 and uL18 co-localize in the nucleolus.
DOI: https://doi.org/10.7554/eLife.31275.009

Flag-uL18 co-immunoprecipitated with HA-MDM2 in a dose-dependent manner, although Myc-SPIN1 itself did not co-immunoprecipitate with HA-MDM2 (*Figure 4B* and *Figure 4—figure supplement 1A*). This effect was specific to the uL18-MDM2 interaction, as Myc-SPIN1 overexpression did not alter the interactions between uL5 and MDM2 (*Figure 4C*). Our immunofluorescence result (*Figure 4D* and *Figure 4—figure supplement 1B*) showed that SPIN1 and uL18 are clearly co-localized in the nucleolus, suggesting that SPIN1 might sequester uL18 in the nucleolus and thus prevent it from binding and inactivating MDM2 in the nucleoplasm. Taken together, these results demonstrate that SPIN1 is a regulator of the uL18-MDM2-p53 pathway, acting by preventing uL18 from interaction with MDM2.

## SPIN1 depletion also causes ribosomal stress, activating p53

Previous studies showed that SPIN1 could recognize H3K4 methylation and stimulate rRNA gene expression, unveiling its role in rRNA synthesis (*Bae et al., 2017*; *Wang et al., 2011*). Disruption of rRNA synthesis leads to disassembly of ribosomal precursors and release of ribosome-free ribosomal proteins from the nucleolus (*Bhat et al., 2004*; *Dai and Lu, 2004*; *Zhang et al., 2003*). Based on these lines of information, we speculated that dysregulation of SPIN1 itself might also impact ribosome biogenesis, resulting in accumulating ribosome-free ribosomal proteins to activate p53. To

test this speculation, we first carried out a sucrose gradient fractionation assay using scramble- and SPIN1-shRNA transfected HCT116[p53+/+] cells. The collected fractions were subjected to western blot (WB) analysis. As anticipated, the levels of uL18 and uL5 in the soluble and ribosome-unbound fractions were markedly increased in SPIN1-depletion cells, accompanying with elevated p53 and MDM2 protein levels (*Figure 5A*). Interestingly, the binding between endogenous uL18/uL5 and MDM2 increased upon SPIN1 knockdown, resembling ribosomal stress (*Figure 5B*). Indeed, as expected, knockdown of SPIN1 reduced the expression of pre-rRNA and rRNA (*Figure 5—figure supplement 1A*).

Moreover, as clearly illustrated in *Figure 5C*, overexpression of SPIN1 compromised p53 activation induced by actinomycin D or 5-Fu treatment, which was reported to trigger ribosomal stress that in turn triggers the formation of RPs-MDM2 complex (*Sun et al., 2007*; *Dai and Lu, 2004*; *Jin et al., 2004*; *Boulon et al., 2010*). In addition, the Y170A mutant of SPIN1, which loses the ability to interact with trimethylated K4 (*Su et al., 2014*; *Wang et al., 2011*), was still able to suppress p53 activity (*Figure 5—figure supplement 1B*), suggesting that SPIN1 regulation of p53 is independent of the activity of SPIN1 in regulating rRNA expression. Our mapping results showed that the critical amino acids Y170, F141 and Y177 for trimethylated K4 interaction (*Su et al., 2014*; *Wang et al., 2011*) are all located in the SPIN1 Tudor two domain that is responsible for uL18 binding (*Figure 5—figure supplement 2A and B*). Tudor two domain truncated mutant of SPIN1 failed to suppress p53 or increase rRNA expression (data not shown). Interestingly, both the N- and C-termini of uL18 were

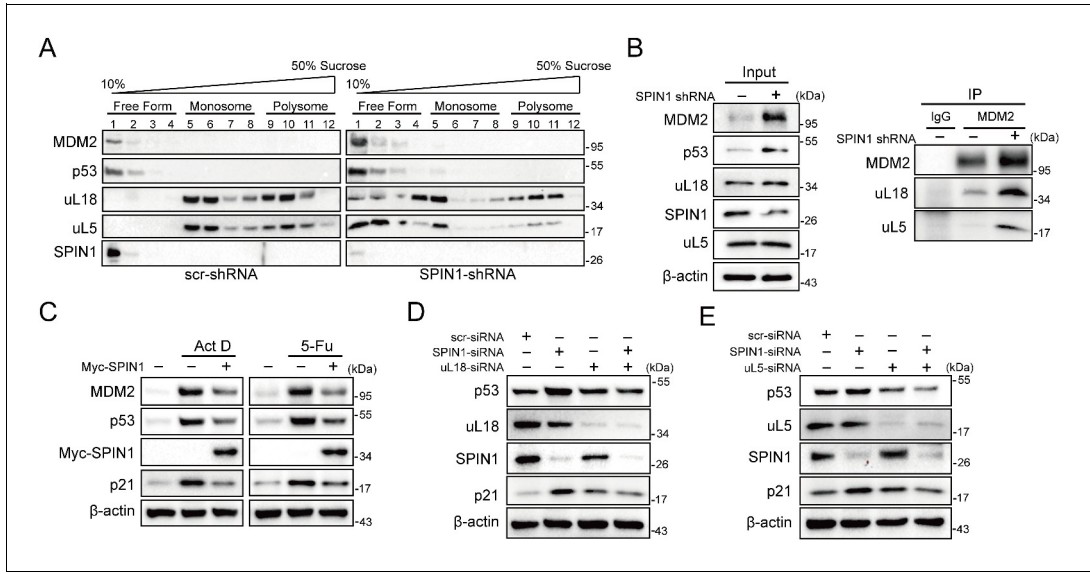

**Figure 5.** SPIN1 depletion increases ribosome-free uL18 and uL5. (**A**) Knockdown of SPIN1 releases free forms of uL18 and uL5. HCT116[p53+/+] were transfected with scramble or SPIN1 shRNA for 36 hr and subjected to sucrose gradient fractionation assay followed by WB analysis with indicated antibodies. (**B**) SPIN1 knockdown increases the endogenous uL18/uL5-MDM2 interaction. Cell lysates of HCT116[p53+/+] cells transfected with scramble or SPIN1 shRNA were immunoprecipitated with MDM2 or control IgG, and analyzed by WB analysis with indicated antibodies. (**C**) SPIN1 overexpression counteracts p53 activation induced by ActD or 5-Fu. U2OS cells were transfected with pcDNA or Flag-SPIN1 for 48 hr, and treated with ActD or 5-Fu for 12 hr before harvested for WB analysis with indicated antibodies. (**D**) and (**E**) Knockdown of uL18 or uL5 compromises the induction of p53 by SPIN1 depletion. U2OS cells were transfected with scramble siRNA, SPIN1 siRNA, uL18 siRNA (**D**) or uL5 siRNA (**E**) as indicated for 48 hr. Cell lysates were subjected to WB analysis with indicated antibodies.
DOI: https://doi.org/10.7554/eLife.31275.010

The following figure supplements are available for figure 5:

**Figure supplement 1.** SPIN1 knockdown reduces rRNA expression and SPIN1-Y170A mutant retains activity to repress p53.
DOI: https://doi.org/10.7554/eLife.31275.011

**Figure supplement 2.** Mapping of domains responsible for uL18-SPIN1 and uL18-MDM2 binding.
DOI: https://doi.org/10.7554/eLife.31275.012

found to bind to SPIN1 (*Figure 5—figure supplement 2C and D*), and these two fragments were required for uL18-MDM2 binding as well (*Figure 5—figure supplement 2E and F*), further supporting our observation that SPIN1 could compete with MDM2 for uL18 binding (*Figure 4B*).

To further confirm the role of these free forms of ribosomal proteins in SPIN1 ablation-induced p53 activation, we knocked down uL18 or uL5 using siRNA with or without SPIN1 depletion in U2OS cells. Strikingly, the reduction of either uL18 or uL5 abrogated SPIN1 knockdown-induced p53 levels, as well as its target p21, as compared to scramble siRNA-transfected cells (*Figure 5D and E*). Collectively, these data indicate that knockdown of SPIN1 could also lead to ribosomal stress, releasing ribosome-free uL18 and uL5, which are required for p53 activation induced by SPIN1 depletion.

## SPIN1 depletion impedes xenograft tumor growth

To translate the above-described cellular functions of SPIN1 into more biological significance, we established a xenograft tumor model by inoculating the aforementioned HCT116 (both p53+/+ and p53-/-) cell lines that expressed scramble shRNA or SPIN1 shRNA into NOD/SCID mice, and monitored tumor size for 18 days. As illustrated in *Figure 6A and B*, SPIN1 knockdown more markedly slowed down the growth of xenograft tumors derived from HCT116$^{p53+/+}$ cells than that from HCT116$^{p53-/-}$ cells. Notably, SPIN1 depletion also reduced the growth of tumors derived from

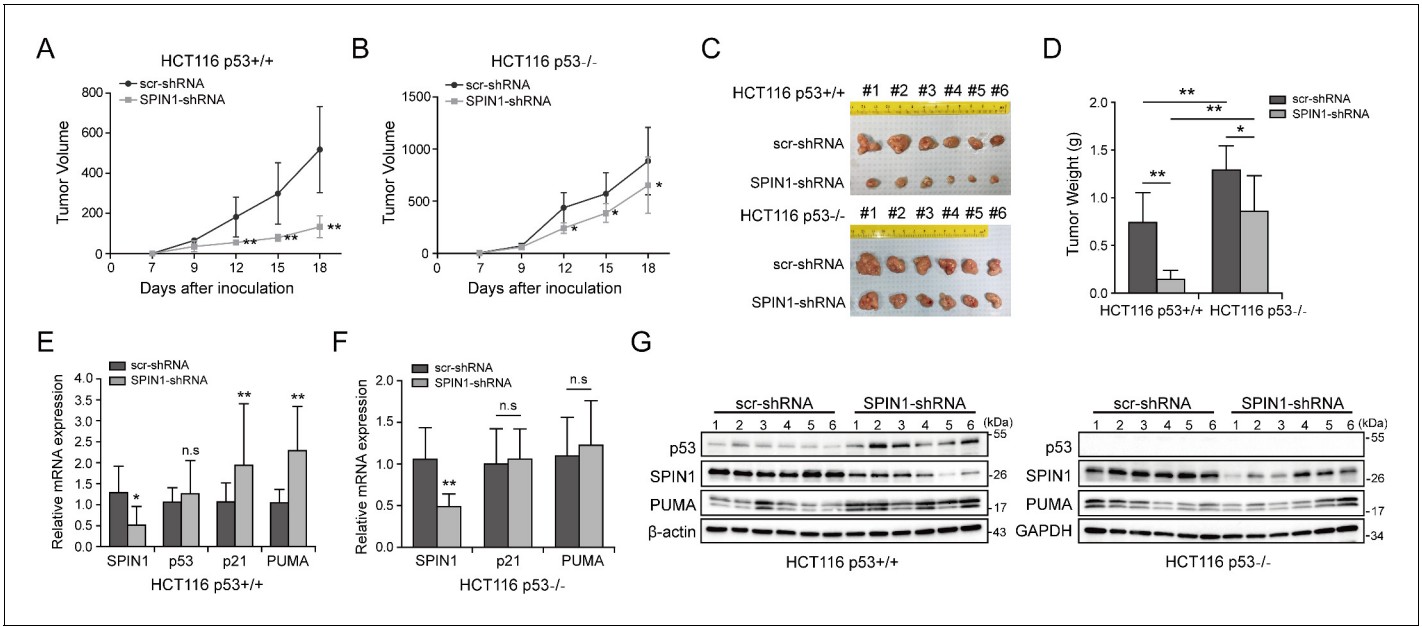

**Figure 6.** SPIN1 knockdown retards tumor growth more dramatically by inducing p53 activity. (**A**) and (**B**) Growth curves of xenograft tumors derived from HCT116$^{p53+/+}$ cells and HCT116$^{p53-/-}$ cells that expressed scramble or SPIN1 shRNA. Data are represented as mean ± SEM, n = 6. (**C**) The images of xenograft tumors that were harvested at the end of experiment. (**D**) Quantification of the average weights of collected tumors from the above experiments. (**E**) and (**F**) The mRNA levels of SPIN1, p53 and p53 target genes were detected in six tumors by RT-qPCR (mean ± SEM, n = 6). (**G**) The protein levels of SPIN1, p53 and p53 targets were detected in six tumors samples by WB analysis with indicated antibodies. *p<0.05, **p<0.01 by two-tailed *t*-test (**D, E, F, G**).

DOI: https://doi.org/10.7554/eLife.31275.013

The following figure supplements are available for figure 6:

**Figure supplement 1.** Quantification of protein expression analyzed from xenograft tumors by Image J software.
DOI: https://doi.org/10.7554/eLife.31275.014

**Figure supplement 2.** High expression of SPIN1 is detected in multiple cancers and associated with poor prognosis in cancer patients.
DOI: https://doi.org/10.7554/eLife.31275.015

**Figure supplement 3.** Western blotting analyses of human colon cancer tissues (n = 22) and normal colon tissue (n = 20) and quantification of SPIN1 expression (Mean ± SEM, p<0.05).
DOI: https://doi.org/10.7554/eLife.31275.016

**Figure supplement 4.** Expression of genes involved in p53 pathway is correlated with SPIN1 expression.
DOI: https://doi.org/10.7554/eLife.31275.017

HCT116$^{p53-/-}$ cells, suggesting that SPIN1 might possess a p53-independent function required for cancer cell growth. In line with the tumor growth curve, the reduction of tumor mass and weight by SPIN1 knockdown was more profound in HCT116$^{p53+/+}$ groups (~60% reduction in weight) than that in HCT116$^{p53-/-}$ groups (~30% reduction in weight) (*Figure 6C and D*). To confirm our cell-based findings, we performed qRT-PCR and WB analysis using the xenograft tumors. As expected, the mRNA levels of p21 and PUMA were significantly increased upon SPIN1 knockdown in HCT116$^{p53+/+}$, but not in HCT116 $^{p53-/-}$ tumors (*Figure 6E and F*). Consistently, the protein levels of p53 and its target PUMA were elevated in HCT116$^{p53+/+}$ groups, but not in HCT116$^{p53-/-}$ groups (*Figure 6G* and *Figure 6—figure supplement 1*). Taken together, these results demonstrate that SPIN1 depletion retards tumor growth by mainly activating p53, although SPIN1 might also possess p53-independent functions in regulation of cell growth and survival.

The data presented above suggest that SPIN1 plays an important role in tumorigenesis. Therefore, we further searched some available genomic and gene expression database for SPIN1 expression in cancers. Interestingly, our analysis of TCGA genome database (*Cerami et al., 2012*; *Gao et al., 2013*) indicated that the SPIN1 gene is markedly amplified in a panel of cancers, including prostate, sarcoma, lung, stomach, breast, head and neck, pancreas and colorectal cancers (*Figure 6—figure supplement 2A*). Consistent with this observation, the analysis of Oncomine database (*Rhodes et al., 2007*) also showed that SPIN1 mRNA expression is extensively upregulated in melanoma tissues when compared with normal skin tissues (~2.367 folds upregulation, *Figure 6—figure supplement 2B*). Moreover, using databases (*Cerami et al., 2012*; *Mizuno et al., 2009*; *SzaszSzász et al., 2016*) that contain gene expression profiles of clinical cancer samples combined with patient outcomes, we found that overexpression of SPIN1 is correlated with poorer prognosis in patients with breast cancer, colorectal cancer and gastric cancer (*Figure 6—figure supplement 2C–F*). Elevated protein expression was also observed in a panel of human colon tumor samples compared with normal tissues (*Figure 6—figure supplement 3*). These data further support that SPIN1 may play an oncogenic role in human cancer progression.

Based on these data and the aforementioned results, we proposed a model for the role of SPIN1 in regulation of p53 (*Figure 7*). Under the condition of low SPIN1 level, nucleolar uL18 escapes from the nucleolus into the nucleoplasm, and works together with uL5 to bind MDM2 and to inhibit its E3 ubiquitin ligase activity toward p53, consequently leading to p53 activation and p53-dependent cell growth arrest and apoptosis, suppressing cancer cell survival (*Figure 7A*). But when SPIN1 levels are high or abnormally elevated in cancer cells, SPIN1 retains uL18 in the nucleolus, thereby preventing uL18 from suppression of MDM2 activity and resulting in p53 degradation, favoring tumor cell growth (*Figure 7B*). This conjecture is further supported by the aforementioned xenograft experiment.

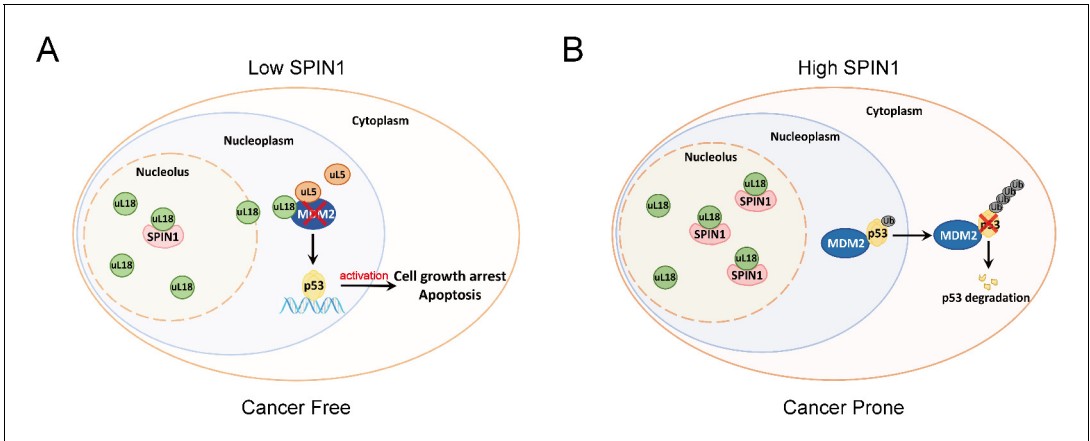

**Figure 7.** A model for SPIN1 regulation of the uL18-MDM2-p53 pathway in cancer.(see text in the Discussion for details).
DOI: https://doi.org/10.7554/eLife.31275.018

## Discussion

The tumor suppressor p53 provides a critical brake on cancer development in response to ribosomal stress, as impairing this ribosomal stress-uL18/uL5-p53 pathway could accelerate tumorigenesis in c-Myc transgenic lymphoma mice (*Macias et al., 2010*). However, it remains largely elusive whether this pathway is subjected to the regulation by other yet unknown proteins. In our attempt to understand molecular insights into this possible regulation, we identified SPIN1, the nucleolar protein important for rRNA synthesis (*Wang et al., 2011*), as a novel regulator of the uL18-MDM2-p53 pathway through interplay with uL18 (*Figure 7*). Our studies as presented here provide the first line of evidence for that SPIN1 acts as an upstream regulator of uL18's accessibility to MDM2 for p53 activation.

Using IP-MS analysis, we identified SPIN1 as a new uL18-associated protein (*Figure 1A*). Our biochemical and cellular experiments using co-IP and GST pull down assays further validated the direct association of SPIN1 with uL18 (*Figure 1B–1D*; *Figure 5—figure supplement 2A–D*). Moreover, we found that SPIN1 and uL18 co-localized in the nucleolus by immunofluorescence assay (*Figure 4D* and *Figure 4—figure supplement 1B*). Remarkably, SPIN1 specifically binds to uL18, but not uL5 and uL14, as no binding was detected between ectopic SPIN1 and uL5 or uL14 by co-IP (*Figure 1E*). Interestingly, SPIN1 does not appear to bind to MDM2, as it was not co-immunoprecipitated with MDM2 either (*Figure 4B* and *Figure 4—figure supplement 1B*). Although our previous reports described a complex of uL18/uL5/uL14-MDM2 (*Dai et al., 2004*), our present findings indicate that SPIN1 may work with uL18 in a separate complex that is different from reported RPs-MDM2 complexes. Also our results suggest that SPIN1 may retain uL18 in the nucleolus so that the latter is unable to shuttle to the nucleoplasm and to inhibit MDM2 activity toward p53 (*Figure 7*).

SPIN1 expression in cells is tightly controlled, as several studies have shown that SPIN1 expression could be negatively monitored by some tumor suppressive non-coding RNAs, such as miR-489 and miR-219–5p (*Chen et al., 2017*; *Drago-Ferrante et al., 2017*; *Li et al., 2017*). Moreover, elevated expression of SPIN1 was strongly correlated with advanced histological stage, chemoresistance and metastasis in patients with breast cancer (*Chen et al., 2016*). Consistent with the aforementioned oncogenic role of SPIN1, our study as presented here showed that SPIN1 depletion by its specific shRNA leads to the augment of the p53-dependent cancer cell growth arrest and apoptosis. This is at least partly because SPIN1 can promote MDM2-dependent ubiquitination and degradation of p53 (*Figure 3*), which is highly likely attributable to its capability to prevent uL18 from binding to MDM2 through retaining uL18 in the nucleolus (*Figure 4*).

Also, knockdown of SPIN1 led to the increase of ribosome-free uL18 and uL5 levels, of the uL18/uL5-MDM2 complex, and of p53 level and activity (*Figure 5A and B*). The activation of p53 by knocking down SPIN1 is due to the ribosomal stress caused by the depletion of this nucleolar protein, as SPIN1 is required for rRNA synthesis by RNA polymerase I (*Wang et al., 2011*). Also, consistent with these observations, overexpression of SPIN1 reduced the activation of p53 by Actinomycin D treatment (*Figure 5C*), whereas knockdown of uL18 or of uL5 impaired the activation of p53 by SPIN1 knockdown (*Figure 5D and E*). Several genes have been implicated to modulate the RPs-MDM2-p53 pathway through interplay with ribosomal proteins (*Sasaki et al., 2011*; *Uchi et al., 2013*; *Kayama et al., 2017*; *Havel et al., 2015*; *Zhang et al., 2014*). In particular, SRSF1 was identified as a component of the RP-MDM2-p53 complex, and could stabilize p53 via uL18 (*Fregoso et al., 2013*). Different from their studies, SPIN1 specifically forms an independent complex with uL18, but not MDM2 or other ribosomal proteins, such as uL5 or uL14, and acts as a negative regulator of p53. Therefore, our present findings unveil a novel mechanism for suppression of the uL18-MDM2-p53 pathway by SPIN1, whose depletion consequently leads to p53-dependent cell growth inhibition and apoptosis.

Consistent with its oncogenic activity, SPIN1 is often amplified in a panel of cancer types with less or no p53 mutation based on our analysis of human samples available in TCGA database (*Figure 6—figure supplement 2A–B*). In addition, elevated SPIN1 expression correlates with poor prognosis in breast, colorectal and gastric cancer patients (*Figure 6—figure supplement 2C–F*). Western blot analysis of a panel of human colon cancer samples revealed that SPIN1 is expressed at significantly higher levels in tumors than in normal tissues (*Figure 6—figure supplement 3*). These findings further indicate that SPIN1 acts as a potential oncogene. In line with these observations, we found that overexpression of SPIN1 promotes cancer cell survival, while knockdown of SPIN1 leads to cancer

cell death as well as the suppression of cancer cell growth and colony formation predominantly in wild-type p53-containing cancer cells (*Figure 2*). Remarkably, knockdown of SPIN1 inhibited xenograft tumorigenesis derived from human colon cancer cells, which was much more significantly in HCT116$^{p53+/+}$ cells than in HCT116$^{p53-/-}$ cells (*Figure 6*). Consistently, bio-informatic analysis on gene expression data of 644 colorectal tumors downloaded from Genomic Data Commons (https://portal.gdc.cancer.gov/) showed that SPIN1 gene expression was correlated with genes (22 genes) enriched in p53 signaling pathway (*Figure 6—figure supplement 4*). These results demonstrate that SPIN1 can promote tumor growth and survival by inactivating p53 and its pathway (*Figure 7*).

Intriguingly, we also found that SPIN1 ablation had a moderate inhibitory effect on cell growth in p53-null HCT116 cells in vitro and in vivo as mentioned above (*Figures 2* and *6*). These findings suggest that SPIN1 must also possess p53-independent oncogenic effects, which might be explained by two possible mechanisms. First, SPIN1 has been reported to execute its oncogenic potentials by activating Wnt and PI3K/Akt pathways (*Chen et al., 2016*; *Wang et al., 2012*), both of which are closely correlated with cancer progression (*Liu et al., 2009*; *Polakis, 2012*). Second, since our previous study has demonstrated that uL18 and uL5 could activate TAp73 through association with MDM2 (*Zhou et al., 2015*), it is possible that the SPIN1-uL18 interaction might impose suppression on TAp73 activity as well, ultimately leading to cell growth arrest and apoptosis.

Recent studies have demonstrated the role of SPIN1 in rRNA transcription (*Bae et al., 2017*; *Su et al., 2014*), which provides a clue that dysregulation of SPIN1 may perturb ribosome biogenesis. In fact, in our current study, we observed that SPIN1 depletion per se increases the levels of ribosome-free uL18 and uL5, accompanying elevated p53 protein levels (*Figure 5*), which recapitulates the effects of ribosomal stress. Our observation that p53 induction caused by SPIN1 depletion could be abrogated by knockdown of either uL18 or uL5 further supports this hypothesis. Therefore, while it is conceivable that SPIN1 counteracts p53 by blocking the interaction between uL18 and MDM2 as discussed above, the mechanism by which disruption of SPIN1 causes ribosomal stress may be also responsible for p53 activation.

In summary, our findings unveil SPIN1 as another novel and important regulator of the MDM2-p53 pathway by predominantly inhibiting the association of uL18 with MDM2 to modulate p53 activity (*Figure 7*) and provide more molecular insights into the fine regulation of this pathway.

# Materials and methods

**Key resources table**

| Reagent type (species) or resource | Designation | Source or reference | Identifiers | Additional information |
|---|---|---|---|---|
| Gene (human) | SPIN1 | National Center for Biotechnology Information (https://www.ncbi.nlm.nih.gov/gene/10927) | Gene ID: 10927; Accession number: NM_006717; UniPro ID: Q9Y657 | |
| Gene (human) | RPL5/uL18 | National Center for Biotechnology Information https://www.ncbi.nlm.nih.gov/gene/6125 | gene ID: 6125; Accession number: NM_000969; UniPro ID: P46777 | |
| Gene (human) | RPL11/uL5 | National Center for Biotechnology Information https://www.ncbi.nlm.nih.gov/gene/6135 | gene ID: 6135; Acctssion number: NM_000975; UniPro ID: P62913 | |
| Gene (human) | RPL23/uL14 | National Center for Biotechnology Information https://www.ncbi.nlm.nih.gov/gene/9349 | gene ID: 9349; Accession number: NM_000978; UniPro ID: P62829 | |
| Gene (human) | TP53 | National Center for Biotechnology Information https://www.ncbi.nlm.nih.gov/gene/7157 | gene ID: 7157; Accession number: NM_000546; UniPro ID: P04637 | |
| Gene (human) | p21/CDKN1A | National Center for Biotechnology Information https://www.ncbi.nlm.nih.gov/gene/1026 | gene ID: 1026; Accession number: NM_000389; UniPro ID: Q42580 | |
| Gene (human) | PUMA/BBC3 | National Center for Biotechnology Information https://www.ncbi.nlm.nih.gov/gene/27113 | gene ID: 27113; Accession number: NM_001127240; UniPro ID: Q9BXH1 | |

*Continued on next page*

*Continued*

| Reagent type (species) or resource | Designation | Source or reference | Identifiers | Additional information |
|---|---|---|---|---|
| Gene (human) | MDM2 | National Center for Biotechnology Information https://www.ncbi.nlm.nih.gov/gene/4193 | gene ID: 4193; Accession number: NM_001145337; UniPro ID: Q00987 | |
| Strain, strain background (mouse) | NOD-SCID | Jackson Laboratories https://www.jax.org/strain/001303 | Stock No: 001303 | |
| Cell line (human) | 293 | ATCC https://www.atcc.org/Products/All/CRL-1573.aspx | Catalog number: ATCC CRL-1573; RRID: CVCL_0045 | |
| Cell line (human) | H1299 | ATCC https://www.atcc.org/Products/All/CRL-5803.aspx | Catalog number: ATCC CRL-5803; RRID: CVCL_0060 | |
| Cell line (human) | U2OS | ATCC https://www.atcc.org/Products/All/HTB-96.aspx | Catalog number: ATCC HTB-96; RRID: RRID:CVCL_0042 | |
| Cell line (human) | H460 | ATCC https://www.atcc.org/Products/All/HTB-177.aspx | Catalog number: ATCC HTB-177; RRID: CVCL_0459 | |
| Cell line (human) | HCT116 p53+/+ | from Dr. Bert Vogelstein at the John Hopkins Medical institutes | | |
| Cell line (human) | HCT116 p53-/- | from Dr. Bert Vogelstein at the John Hopkins Medical institutes | | |
| Cell line (human) | MEF (Mdm2-/-; p53-/-) | from Dr. Guillermina Lozano from MD Anderson Cancer Center, the University of Texas. | | |
| Antibody | Mouse anti-human Flag monoclonal antibody | Sigma-Aldrich | Catalog number: F1804; RRID: AB_262044 | Applications: WB; Immunofluoresce |
| Antibody | Mouse anti-human Myc monoclonal antibody | Santa Cruz Technology | Catalogue number: sc-40 | Applications: WB; Immunofluoresce |
| Antibody | Mouse anti-human GFP monoclonal antibody | Santa Cruz Technology | Catalogue number: sc-9996; RRID: AB_627695 | Applications: WB; Immunofluoresce |
| Antibody | Mouse anti-human GST monoclonal | ProteinTech | Catalogue number: HRP-66001; RRID: AB_10951482 | Applications: WB |
| Antibody | Rabbit anti-bacterial His polyclonal antibody | ProteinTech | Catalogue number: 10560–1-lg; RRID: AB_1607770 | Applications: WB |
| Antibody | Rabbit anti-human SPIN1 polyclonal antibody | ProteinTech | Catalogue number: 12105–1-AP; RRID: AB_2196111 | Applications: WB |
| Antibody | Mouse anti-human p53 monoclonal antibody | Santa Cruz Technology | Catalogue number: sc-126; RRID: AB_628082 | Applications: WB |
| Antibody | Mouse anti-human p21 monoclonal antibody | Neomarkers, Fremont, | Catalogue number: MS-891-P0; RRID: AB_143907 | Applications: WB |
| Antibody | Rabbit anti-human PUMA polyclonal antibody | ProteinTech | Catalogue number: 55120–1-AP; RRID: AB_10859944 | Applications: WB |
| Antibody | Mouse anti-human β-actin monoclonal antibody | Santa Cruz Technology | Catalogue number: sc-47778; RRID: AB_2714189 | Applications: WB |
| Antibody | Rabbit anti-human GAPDH polyclonal antibody | Proteintech | Catalogue number: 10494–1-AP; RRID: AB_2263076 | Applications: WB |

*Continued on next page*

*Continued*

| Reagent type (species) or resource | Designation | Source or reference | Identifiers | Additional information |
|---|---|---|---|---|
| Chemical compound, drug | Cycloheximide | Sigma-Aldrich | Catalogue number: 66-81-9 | |
| Chemical compound, drug | MG-132 | Sigma-Aldrich | Catalogue number: 474787 | |
| Chemical compound, drug | 5-FU | Sigma-Aldrich | Catalogue number: 51218 | |
| Chemical compound, drug | Actinomycin D (Act D) | Sigma-Aldrich | Catalogue number: 50-76-0 | |

## Cell culture and transient transfection

U2OS, H1299, HEK293 and H460 cells were purchased from American Type Culture Collection (ATCC). HCT116$^{P53+/+}$ and HCT116$^{P53-/-}$ cells were generous gifts from Dr. Bert Vogelstein at the John Hopkins Medical institutes. MEF$^{p53-/-;Mdm2-/-}$ cells were generous gifts from Dr. Guillermina Lozano from MD Anderson Cancer Center, the University of Texas. STR profiling was performed to ensure cell identity. No mycoplasma contamination was found. All cells were cultured in Dulbecco's modified Eagle's medium (DMEM) supplemented with 10% fetal bovine serum, 50 U/ml penicillin and 0.1 mg/ml streptomycin and were maintained at 37°C in a 5% $CO_2$ humidified atmosphere. Cells were seeded on the plate the day before transfection and then transfected with plasmids as indicated in figure legends using TurboFect transfection reagent according to the manufacturer's protocol (Thermo Scientific, R0532). Cells were harvested at 30–48 hr post-transfection for future experiments.

## Plasmids and antibodies

The Myc-tagged SPIN1 plasmid was generated by inserting the full-length cDNA amplified by PCR into the pcDNA3.1/Myc-His vector at EcoR I and Bam HI, using the following primers, forward-CGGAATTCatgaagaccccattcggaaag; reverse-CGGGATCCggatgtttttcaccaaaatcgtag. Flag-SPIN1 was generated by inserting SPIN1 cDNA into 2Flag-pcDNA3 at BamHI and XhoI sites. The primers used for PCR amplifying reverse transcribed mRNA were: forward-CGGGATCCaagaccccattcggaaagaca; reverse-CCGCTCGAGctaggatgtttttcaccaaatcgta. The GST-tagged SPIN1 fragments, His-tagged SPIN1, GFP-tagged SPIN1 and FLAG-tagged SPIN1-Y170A plasmids were generous gifts from Drs. Bing Zhu from Institute of Biophysics, Chinese Academy of Sciences, and Haitao Li from Tsinghua University, Beijing, China. The plasmids SPIN1 shRNA-1 and −2 were purchased from Sigma-Aldrich (St Louis, MO). The plasmids encoding HA-MDM2, Flag-uL18, Flag-uL5, Flag-uL14, GFP-uL18, p53, His-Ub, GST-MDM2, His-uL18 and GST-uL18 fragments were described previously (*Dai et al., 2004*; *Dai and Lu, 2004*). Anti-Flag (Sigma-Aldrich, catalogue no. F1804, diluted 1:3000), anti-Myc (9E10, Santa Cruz Technology, catalogue no. sc-40, diluted 1:1000), anti-GFP (B-2, Santa Cruz Technology, catalogue no.sc-9996, diluted 1:1000), anti-SPIN1 (Proteintech, Rosemont, IL, USA catalogue no. 12105–1-AP), anti-p53 (DO-1, Santa Cruz Technology, catalogue no. sc-126, diluted 1:1000), anti-p21 (CP74, Neomarkers, Fremont, catalogue no. MS-891-P0, diluted 1:1000), anti-PUMA (Proteintech, catalogue no. 55120–1-AP), anti-β-actin (C4, Santa Cruz Technology, catalogue no.sc-47778, diluted 1:5000), anti-GAPDH (Proteintech, catalogue no. 10494–1-AP), were commercially purchased. Antibodies against MDM2 (2A9 and 4B11), uL18 and uL5 were described previously (*Dai et al., 2004*; *Dai and Lu, 2004*).

## GST fusion protein-protein interaction assay

GST-tagged SPIN1 or GST-tagged uL18 fragments were expressed in *E. coli* and conjugated with glutathione-Sepharose 4B beads (Sigma-Aldrich). His-tagged SPIN1 and His-tagged uL18 were purified using a Ni-NTA (QIAGEN, Valencia, CA, USA) column, and eluted with 0.5 M imidazole. Protein-protein interaction assays were conducted as described previously (*Jin et al., 2002*). Briefly, for *Figure 6A*, 500 ng of purified His-tagged uL18 protein were incubated and gently rotated with the glutathione-Sepharose 4B beads containing 300 ng of GST-SPIN1 fragments or GST only at 4°C for 4 hr. For *Figure 6C*, 300 ng of purified His-tagged SPIN1 protein were incubated and gently shaked

with the glutathione-Sepharose 4B beads containing 200 ng of GST-uL18 fragments or GST only at 4°C for 1 hr. The mixtures were washed three times with GST lysis buffer (50 mM Tris/HCT pH 8.0, 0.5% NP-40, 1 mM EDTA, 150 mM NaCl, 10% glycerol). Bound proteins were analyzed by IB with the antibodies as indicated in the figure legends.

## Reverse transcription (RT) and quantitative RT-PCR analysis

Total RNA was isolated from cells or tissues using Trizol (Invitrogen, Carlsbad, CA) following the manufacturer's protocol. Total RNAs of 0.5 or 1.0 µg were used as template for reverse transcription using poly-(T)20 primers and M-MLV reverse transcriptase (Promega, Madision, WI). Quantitative RT-PCR (RT-PCR) was performed using SYBR Green Mix following the manufacturer's protocol (BioRad, Hercules, CA, USA). The primers for SPIN1, p53, p21, PUMA, pre-rRNA, 18S rRNA, rRNA, and GAPDH cDNA are as follows: SPIN1, 5'-CAGAGCTGATGCAGGCCAT-3' and 5'-ACTGGGTAA-CAGGGCCATTG-3', p53, 5'-CCCAAGCAATGGATGATTTGA-3' and 5'-GGCATTCTGGGAGCTTCA TCT-3'; p21, 5'-CTGGACTGTTTTCTCTCGGCTC-3' and 5'-TGTATATTCAGCATTGTGGGAGGA-3'; PUMA, 5'-ACAGTACGAGCGGCGGAGACAA-3' and 5'-GGCGGGTGCAGGCACCTAATT-3'; pre-rRNA, 5'-GCTCTACCTTACCTACCTGG-3' and 5'-TGAGCCATTCGCAGTTTCAC-3'; 18S rRNA, 5'-GCTTAATTTGACTCAACACGGGC-3' and 5'-AGCTATCAATCTGTCAATCCTGTC-3'; rRNA, 5'-TGA-GAAGACGGTCGAACTTG-3' and 5'-TCCGGGCTCCGTTAATGATC-3'; GAPDH, 5'-GATTCCACCCA TGGCAAATTC-3' and 5'-AGCATCGCCCCACTTGATT-3'.

## Flow cytometry analysis

Cell transfected with scramble shRNA or SPIN1 shRNAs as indicated in the figure legends were fixed with 70% ethanol overnight and stained in 500 µl of propidium iodide (PI, Sigma-Aldrich) stain buffer (50 µg/ml PI, 200 µg/ml RNase A, 0.1% Triton X-100 in phosphate-buffered saline) at 37°C for 30 min. The cells were then analyzed for DNA content using a BD Biosciences FACScan flow cytometer (BD Biosciences, San Jose, CA). Data were analyzed using the CellQuest (BD Biosciences) and Modfit (Verity, Topsham, ME) software programs.

## Annexin V assay

Cells transfected with scramble shRNA or SPIN1 shRNA were split into 96-well plate and IncuCyte Annexin V Green Reagent for apoptosis was added to each well at the time of seeding. Cell apopto-sis was monitored using IncuCyte S3 live-cell imaging system.

## Cell viability assay

To assess the long-term cell survival, the Cell Counting Kit-8 (CCK-8) (Dojindo Molecular Technolo-gies, Rockville, MD) was used according to the manufacturer's instructions. Cell suspensions were seeded at 2000 cells per well in 96-well culture plates at 12 hr post-transfection. Cell viability was determined by adding WST-8 at a final concentration of 10% to each well, and the absorbance of these samples was measured at 450 nm using a Microplate Reader (Molecular Device, SpecrtraMax M5e, Sunnyvale, CA) every 24 hr for 5 days.

## Colony formation assay

Cells were trypsinized and seeded at equal number of cells on 60 mm plates. Media were changed every 4 days until the colonies were visible. Puromycin was added into the media for selection at a concentration of 2 µg/ml. Cells were fixed with methanol and stained with crystal violet solution at RT for 30 min. ImageJ was used for quantification of the colonies.

## Western blot analysis

Cells were harvested and lysed in lysis buffer consisting of 50 mM Tris/HCl (pH 7.5), 0.5% Nonidet P-40 (NP-40), 1 mM EDTA, 150 mM NaCl, 1 mM dithiothreitol (DTT), 0.2 mM phenylmethylsulfonyl fluoride (PMSF), 10 µM pepstatin A and 1 mM leupeptin. Equal amounts of clear cell lysates (20–80 µg) were used for WB analysis as described previously (*Chao et al., 2016*; *Zhou et al., 2016*). Human samples originally obtained from Indiana University Simon Cancer Center Solid Tissues Bank were ground and lysed in lysis buffer before western blot analysis.

## In vivo ubiquitination assay

HCT116$^{p53-/-}$ cells were transfected with plasmids encoding p53, HA-MDM2, His-Ub or Myc-SPIN1 as indicated in the figure legends. At 48 hr after transfection, cells were harvested and split into two aliquots, one for WB analysis and the other for ubiquitination assay. Briefly, cell pellets were lysed in buffer I (6 M guanidinium-HCT, 0.1 M Na$_2$HPO$_4$/NaH$_2$PO$_4$, 10 mM Tris-HCl (pH 8.0), 10 mM β-mercaptoethanol) and incubated with Ni-NTA beads (Qiagen) at room temperature for 4 hr. Beads were washed once with buffer I, buffer II (8 M urea, 0.1 M Na$_2$HPO$_4$/NaH$_2$PO$_4$, 10 mM Tris-HCl (pH 8.0), 10 mM β-mercaptoethanol), and buffer III (8 M urea, 0.1 M Na$_2$HPO$_4$/NaH$_2$PO$_4$, 10 mM Tris-HCl (pH 6.3), 10 mM β-mercaptoethanol). Proteins were eluted from beads in buffer IV (200 mM imidazole, 0.15 M Tris-HCl (pH 6.7), 30% glycerol, 0.72 M β-mercaptoethanol, and 5% SDS). Eluted proteins were analyzed by WB with indicated antibodies as previously reported (*Zhou et al., 2016*).

## Immunoprecipitation

Immunoprecipitation (IP) was conducted using antibodies as indicated in the figure legends. Briefly, ~500–1000 μg of proteins were incubated with the indicated antibody at 4°C for 4 hr or overnight. Protein A or G beads (Santa Cruz Biotechnology) were then added, and the mixture was incubated at 4°C for additional 1 to 2 hr. Beads were wash at least three times with lysis buffer. Bound proteins were detected by WB analysis with antibodies as indicated in the figure legends.

## RNA interference

SiRNAs against SPIN1, uL18 and uL5 were commercially purchased from Ambion. SiRNAs (20–40 nm) were introduced into cells using TurboFect transfection reagent following the manufacturer's instruction. Cells were harvested 48–72 hr post-transfection for WB or RT-PCR.

## Immunofluorescence staining

Cells were fixed in 4% paraformaldehyde (PFA) for 25 min, followed by permeabilization in 0.3% Triton X-100 for 20 min. The fixed cells were blocked with 5% bovine serum albumin for 30 min, and then the cells were incubated with indicated antibodies at 4°C overnight. Cells were then washed and incubated with the corresponding secondary antibody and 4′−6-diamidino-2-phenylindole (DAPI) for nuclear staining. The cellular localization of SPIN1 or uL18 was examined under a confocal microscope (Nikon, ECLIPSE Ti2).

## Sucrose gradient fractionation and ribosome profiling

This assay was performed following the protocol previously described (*Guo et al., 2010*). Briefly, cells were harvested at 70–80% confluence after halting translation by 100 μg/ml cycloheximide incubation for 10 min. Cells were lysed in lysis buffer (10 mM Tris-HCl (pH 7.4), 5 mM MgCl$_2$, 100 mM KCl, 1% Triton X-100) and gently sheared with a 26-gauge needle for four times. Lysates were subjected to 10–50% sucrose gradient centrifugation and the fractions were collected through BR-188 Density Gradient Fractionation System (Brandel, Gaithersburg, MD).

## Generating stable cell lines

Briefly, scramble shRNA or SPIN1 shRNAs purchased from Sigma were transfected into HCT116$^{p53+/+}$ and HCT116$^{p53-/-}$ cells using TurboFect reagent. The cells were maintained at 37°C in a 5% CO$_2$ humidified atmosphere for 48 hr and were split to two aliquots, one for WB analysis and the other for selection using final concentration of 2 μg/ml puromycin in growth medium.

## Mouse xenograft experiments

Seven-week-old female NOD/SCID mice were purchased from Jackson Laboratories. Mice were randomized into two groups (six mice in each) and subcutaneously inoculated with 5 × 10$^6$ HCT116 cells that stably expressing scramble shRNA or SPIN1 shRNA in the right and left flanks, respectively. Tumor growth was monitored every other day with electronic digital calipers (Thermo Scientific) in two dimensions. Tumor volume was calculated with the formula: tumor volume (mm$^3$) = (length-×width$^2$)/2. Mice were sacrificed by euthanasia, and tumors were harvested and weighed. To detect p53 activation and apoptosis in vivo, the RNAs and proteins were disrupted from tumors via homogenization in Trizol or RIPA buffer, and then subjected to RT-qPCR and WB analysis. The experiment

was not blind and was handled according to approved institutional animal care and use committee (IACUC) protocol (#4275R) of Tulane University School of Medicine. The maximum tumor volume per tumor allowed the IACUC committee is 1.5 cm diameter or 300 mm$^3$ per tumor.

## TCGA data analysis

From Genomic Data Commons (https://portal.gdc.cancer.gov/), we downloaded the digital gene expression data of 644 colorectal cancer tumors, which was generated using a RNA-seq platform by the Cancer Genome Atlas (TCGA). In the data set, gene expression levels were measured with FPKM (Fragments Per Kilobase of transcript per Million mapped reads) and normalized using the Upper Quantile method.

We condensed the data by excluding the genes that were not expressed in over 75% samples. Logarithm transformation was applied to the expression levels of the remaining ~28,700 Ensembl genes. The transcriptional correlations between the SPIN1 gene and the others genes were evaluated using Pearson correlation coefficient (r). The corresponding p-values were estimated by t-tests. On the cutoffs, including the absolute value of r being larger than 0.3 and Bonferroni adjusted p-value being less than 0.01, ~4500 significant genes were selected. The functional enrichment test of the selected genes was performed using the DAVID tool (*Huang et al., 2009*) (https://david.ncifcrf.gov/). The heatmap was generated using R function, heatmap.2().

## Statistical testing

All in vitro experiments were performed in biological triplicate and reproduced at least twice. The Student's two-tailed t-test was used to determine mean difference among groups. $p < 0.05$ was considered statistically significant, asterisks represent significance in the following way: $*p < 0.05$; $**p < 0.01$. The term 'n.s' indicates that no significant difference was found. All the data are presented as mean ± SEM.

## Acknowledgements

We thank Drs. Bert Vogelstein and Guillermina Lozano for offering HCT116 cells and MEF cells, respectively, Drs. Bing Zhu and Haitao Li for offering plasmids, the Lu lab members and Dr. Hee-Won Park for active discussion and suggestions. Hua Lu and Shelya X Zeng were supported in part by NIH-NCI grants R01CA095441, R01CA172468, R01CA127724, R21CA190775, and R21 CA201889. Kun Zhang was supported in part by NIH grant 2G12MD007595.

## Additional information

### Funding

| Funder | Grant reference number | Author |
|---|---|---|
| National Institutes of Health | 2G12MD007595 | Kun Zhang |
| National Institutes of Health | R01CA095441 | Shelya X Zeng<br>Hua Lu |
| National Institutes of Health | R01CA172468 | Hua Lu<br>Shelya X Zeng |
| National Institutes of Health | R01CA127724 | Hua Lu<br>Shelya X Zeng |
| National Institutes of Health | R21CA190775 | Hua Lu<br>Shelya X Zeng |
| National Institutes of Health | R21 CA201889 | Hua Lu<br>Shelya X Zeng |
| National Cancer Institute | | Hua Lu |

The funders had no role in study design, data collection and interpretation, or the decision to submit the work for publication.

## Author contributions

Ziling Fang, Conceptualization, Data curation, Formal analysis, Validation, Investigation, Visualization, Methodology, Writing—original draft, Writing—review and editing; Bo Cao, Conceptualization, Data curation, Formal analysis, Supervision, Validation, Investigation, Visualization, Methodology, Writing—original draft, Writing—review and editing; Jun-Ming Liao, Conceptualization, Data curation, Formal analysis, Supervision, Validation, Investigation, Visualization, Methodology, Writing—review and editing; Jun Deng, Data curation, Formal analysis, Validation, Investigation, Visualization, Methodology; Kevin D Plummer, Data curation, Formal analysis, Investigation, Methodology; Peng Liao, Data curation, Formal analysis, Supervision, Investigation; Tao Liu, Investigation, Assisting Dr. Ziling Fang in performing some cellular experiments; Wensheng Zhang, Kun Zhang, Data curation, Software; Li Li, Resources, Visualization, colon cancer specimen analysis; David Margolin, Resources, Validation, Offering colon cancer specimens; Shelya X Zeng, Conceptualization, Resources, Data curation, Formal analysis, Supervision, Funding acquisition, Investigation, Methodology, Project administration, Writing—review and editing; Jianping Xiong, Resources, Supervision, Project administration, Writing—review and editing; Hua Lu, Conceptualization, Resources, Data curation, Formal analysis, Supervision, Funding acquisition, Validation, Investigation, Writing—original draft, Project administration, Writing—review and editing

## Author ORCIDs

Hua Lu (iD) http://orcid.org/0000-0002-9285-7209

## Ethics

Animal experimentation: The experiment was not blind and was handled according to approved institutional animal care and use committee (IACUC) protocol (#4275R) of Tulane University School of Medicine. The maximum tumor volume per tumor allowed the IACUC committee is 1.5 cm diameter or 300 mm3 per tumor.

## Decision letter and Author response

Decision letter https://doi.org/10.7554/eLife.31275.022
Author response https://doi.org/10.7554/eLife.31275.023

## Additional files

### Supplementary files

• Transparent reporting form
DOI: https://doi.org/10.7554/eLife.31275.019

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
