## [Decision Letter]

Thank you for submitting your article "SPIN1 promotes tumorigenesis by blocking the uL18-MDM2-p53 pathway" for consideration by *eLife*. Your article has been favorably evaluated by James Manley (Senior Editor) and three reviewers, one of whom is a member of our Board of Reviewing Editors. The reviewers have opted to remain anonymous.

The reviewers have discussed the reviews with one another and the Reviewing Editor has drafted this decision to help you prepare a revised submission.

Summary:

Previous studies from multiple laboratories including the authors' have implicated specific ribosomal proteins in the regulation of the tumor suppressor p53 via effects on its negative regulator MDM2. In particular, RPL5/uL18 has been shown to inhibit the ability of MDM2 to target p53 for degradation. In this manuscript, the authors present data to argue that SPIN1 interacts with uL18 thereby preventing its effects on MDM2. Since SPIN1 can be shown to be upregulated in a subset of human cancers, the authors conclude that SPIN1 promotes tumorigenesis via these effects. The manuscript is clearly written, the data are rigorously presented, and the conclusions are reasonable and justified. Elucidation of how the tumor suppressor p53 is regulated in response to different cellular signaling pathways is a significant area. The authors show:

1) SPIN1 knockdown stabilizes and activates p53 signaling. They also show that overexpression of SPIN1 increases MDM2-mediated degradation of p53, and thus dampens down the p53 response.

2) Mechanistically the authors find that SPIN1 regulates p53-MDM2 by sequestering uL18 in the nucleolus, so that uL18 cannot bind and inhibit MDM2; thus silencing SPIN1 leads to p53 stabilization.

3) The authors show that SPIN1 overexpression may enhance tumor growth, as in the Hct p53 WT and -/- pair, shRNA for SPIN1 impedes the growth of tumor xenografts in a partially p53-dependent manner.

Overall, the work is technically adept and done in multiple cell lines. The data presented are of high quality and support the conclusions of the manuscript. Finally, the manuscript is well written and will be of interest to the p53 and ribosome stress fields, provided the following issues are adequately addressed.

Essential issues to address:

1) In this reviewer’s opinion, the data in Figure 6—figure supplement 2 add significantly to the interesting nature of the manuscript, and this figure should be in the main figures of the paper. In the end, however, we must be cautious about RNA overexpression data, such as from TCGA and other databases. The manuscript would be improved if the authors were to validate the overexertion of SPIN1 in tumors by tumor micro-array of one of the selected tumor types, to show that SPIN1 is indeed overexpressed at the protein level in one of these tumor types. The authors could then use this opportunity, if they chose, to see whether SPIN1 level correlated with tumor grade.

2) In the end, although the authors show that overexpressed SPIN1 can dampen down the p53 pathway, the authors themselves show that silencing SPIN1 also reduces tumor growth in a p53-independent manner. The authors mention that SPIN1 can also impact the Wnt and AKT pathways. So the reader is left wondering about the relevance of the p53, versus these other pathways, with regard to the selection for SPIN1 overexpression that occurs in human cancer. To address this issue of relevance, the data in Figure 6—figure supplement 2 support the following, testable hypothesis: the overexpression of SPIN1 should correlate with decreased expression of p53 pathway genes (if this is indeed the reason tumors overexpress this protein). A bio-informatic analysis of p53 pathway genes, versus Wnt-pathway associated genes, in colorectal or gastric tumors (or whatever tumor type the authors prefer) with high versus low SPIN1 would add significantly to the take home message of this manuscript, and might even shed light as to which pathway tumors are selecting for, when they amplify or overexpress SPIN1. This would also alleviate one of the weaknesses of the manuscript, as it would highlight the consequences of SPIN1 overexpression in cancer. A good bio-informaticist could do these studies in a relatively short amount of time, and these findings would markedly strengthen the manuscript.

3) The authors present extensive data to support the notion of an interaction between SPIN1 and uL18. Additionally, they show evidence that suggests that SPIN1 can regulate ribosome integrity, and thus SPIN1 could thereby influence p53 activity via this distinct mechanism. This latter notion is underdeveloped in the manuscript. Along these lines, it is unclear what is the relative contributions of each of these mechanisms to the regulation of the p53-MDM2 interplay. This premise should be clarified with additional experimental evidence. For example, the authors could determine whether SPIN1 mutants that lose binding to uL18 still have effects on rRNA levels. This would confirm that the effect of SPIN1 on p53 and the ribosomal checkpoint are distinct. Or the authors could use the binding mutants identified in biological assays toward this question.

4) With regard to Figure 3: To support the authors' hypothesis, it would be important to show here that ectopic SPIN1-mediated p53 ubiquitination is blocked by knocking down uL18. (This experiment was done in the absence of proteasome inhibitor MG132. Can the authors comment as to why p53 levels are not increasingly diminished?)

5) Figure 4: ectopic SPIN1 only modestly – not 'markedly' – reduces the ectopic uL18-mediated p53 protein and MDM2 gene induction. The p21 effect is however, more significant. Can the authors look at other p53 target genes and show how these respond?

6) Model Figure 6: The authors should exclude that SRSF1 – which also binds to uL18 and activates p53 – plays any role in the SPIN1/uL18 complex.

7) Figure 6—figure supplement 2: With the exception of NEPC (what is it – Neuroendocrine prostate cancer?) tumors, the alteration frequency of SPIN1 in this large cancer panel is very low (and includes deletions) – and is likely not above the general mutational noise seen in cancers. How does Figure 6—figure supplement 2 look when only wt p53 tumors are selected?

---

## [Author Response]

Essential issues to address:1) In this reviewer’s opinion, the data in Figure 6—figure supplement 2 add significantly to the interesting nature of the manuscript, and this figure should be in the main figures of the paper. In the end, however, we must be cautious about RNA overexpression data, such as from TCGA and other databases. The manuscript would be improved if the authors were to validate the overexertion of SPIN1 in tumors by tumor micro-array of one of the selected tumor types, to show that SPIN1 is indeed overexpressed at the protein level in one of these tumor types. The authors could then use this opportunity, if they chose, to see whether SPIN1 level correlated with tumor grade.

We thank the reviewer for the comments. As the data from Figure 6—figure supplement 2) were all derived from online genomics database, and some of them have limited sample numbers as the reviewer pointed out elsewhere, we believe it is more appropriate to use the data as a supplementary figure.

We agree with the reviewer that RNA data and other databases need to be validated. Unfortunately, the SPIN1 antibody is not suitable for immunohistochemistry analysis, because of high background staining. Alternatively, we performed Western blot analysis on a panel of human tissues (20 normal vs. 22 colon tumor) and found that SPIN1 is indeed significantly higher in tumors than that in normal tissues. The result is included as Figure 6—figure supplement 3in the revised manuscript.

2) In the end, although the authors show that overexpressed SPIN1 can dampen down the p53 pathway, the authors themselves show that silencing SPIN1 also reduces tumor growth in a p53-independent manner. The authors mention that SPIN1 can also impact the Wnt and AKT pathways. So the reader is left wondering about the relevance of the p53, versus these other pathways, with regard to the selection for SPIN1 overexpression that occurs in human cancer. To address this issue of relevance, the data in Figure 6—figure supplement 2 support the following, testable hypothesis: the overexpression of SPIN1 should correlate with decreased expression of p53 pathway genes (if this is indeed the reason tumors overexpress this protein). A bio-informatic analysis of p53 pathway genes, versus Wnt-pathway associated genes, in colorectal or gastric tumors (or whatever tumor type the authors prefer) with high versus low SPIN1 would add significantly to the take home message of this manuscript, and might even shed light as to which pathway tumors are selecting for, when they amplify or overexpress SPIN1. This would also alleviate one of the weaknesses of the manuscript, as it would highlight the consequences of SPIN1 overexpression in cancer. A good bio-informaticist could do these studies in a relatively short amount of time, and these findings would markedly strengthen the manuscript.

We thank the reviewer for the comments. We performed bio-informatic analysis (details in revised manuscript) on the gene expression data of 644 colorectal cancer tumors downloaded from Genomic Data Commons (https://portal.gdc.cancer.gov/) and found that SPIN1 gene expression was correlated with genes enriched in p53 signaling pathway (Figure 6—figure supplement 4). However, no significant correlation between SPIN1 and WNT signaling pathway was observed (data not shown).

3) The authors present extensive data to support the notion of an interaction between SPIN1 and uL18. Additionally, they show evidence that suggests that SPIN1 can regulate ribosome integrity, and thus SPIN1 could thereby influence p53 activity via this distinct mechanism. This latter notion is underdeveloped in the manuscript. Along these lines, it is unclear what is the relative contributions of each of these mechanisms to the regulation of the p53-MDM2 interplay. This premise should be clarified with additional experimental evidence. For example, the authors could determine whether SPIN1 mutants that lose binding to uL18 still have effects on rRNA levels. This would confirm that the effect of SPIN1 on p53 and the ribosomal checkpoint are distinct. Or the authors could use the binding mutants identified in biological assays toward this question.

We thank the reviewer for the comments. We generated Tudor 2 domain deletion mutation construct of SPIN1 (dT2), which loses binding to uL18 as shown in our manuscript. This mutant failed to suppress p53. However, this mutant also failed to regulate rRNA expression as shown in Author response image 1, as Y170, F141 and Y177, the most critical amino acids for interaction with trimethylated K4 are all located in the Tudor 2 domain (AA132-193) (Su et al., 2014; Wang et al., 2011). Therefore, alternatively, we tested the effect of SPIN1-Y170A mutant on p53. Interestingly, this mutant was still able to inhibit p53, suggesting that SPIN1 regulation of p53 is independent of the activity of SPIN1 in regulating rRNA expression (Figure 5—figure supplement 1).

**Author response image 1. respfig1:** SPIN1-dT2 fails to inhibit p53. U2OS cells were transfected with wild type and Tudor 2 domain deletion mutant (FLAG-dT2) SPIN1 and cells were harvested 48 hrs after transfection for Western Blot (Left panel) and qRT-PCR (Right panel) analyses.

4) With regard to Figure 3: To support the authors' hypothesis, it would be important to show here that ectopic SPIN1-mediated p53 ubiquitination is blocked by knocking down uL18. (This experiment was done in the absence of proteasome inhibitor MG132. Can the authors comment as to why p53 levels are not increasingly diminished?)

uL18 has been shown to inhibit MDM2-mediated p53 ubiquitination and knockdown of uL18 attenuates ribosomal stress (low dose of Actinomycin D) induction of p53 (Dai et al., 2004). It is therefore less likely to observe blocked SPIN1-mediated p53 ubiquitination by knocking down uL18 according to our proposed mechanism, in which, SPIN1 binds to and sequesters uL18 in the nucleolus to prevent uL18 from binding to MDM2, resulting in similar effect on MDM2-mediated p53 ubiquitination to what knocking down uL18 would do.

The experiment was done in the presence of MG132 as described in the figure legend.

5) Figure 4: ectopic SPIN1 only modestly – not 'markedly' – reduces the ectopic uL18-mediated p53 protein and MDM2 gene induction. The p21 effect is however, more significant. Can the authors look at other p53 target genes and show how these respond?

We thank the reviewer for the comments. We now modified the expression accordingly. We also included Western blot result of another p53 target, PUMA, in the revised manuscript.

6) Model Figure 6: The authors should exclude that SRSF1 – which also binds to uL18 and activates p53 – plays any role in the SPIN1/uL18 complex.

It is intriguing to bring out other proteins, such as SRSF1, possibly involved in the uL18/p53 pathway. To our anticipation, it is less likely that SRSF1 could participate in SPIN1 regulation of uL18. From the literature, SRSF1 forms a complex with uL18 and MDM2, which primarily occurs in the nucleoplasm, in response to ribosomal stress, resulting in p53 activation (Fregoso et al., 2013). By contrast, as shown in our current study (Figure 1–Figure 7), SPIN1 is a nucleolar protein that does not bind to MDM2 directly, but can sequester uL18 in the nucleolus, keeping uL18 from suppressing MDM2 and leading to p53 inhibition. It is possible that SPIN1 might reduce the interaction between of SRSF1 and uL18 since uL18 can be retained in the nucleolus by SPIN1, similar to the way SPIN1 reduces the uL18-MDM2 complex. However, addressing this issue as well as other more related issues that will be certainly brought up when we work on the functional interplay between SRSF1 and SPIN1 in the uL18-p53 pathway should be an independent research topic and would generate more substantial amounts of new data for another possible manuscript. We hope that this reviewer would agree with us and allow us to address this question in our future study.

7) Figure 6—figure supplement 2: With the exception of NEPC (what is it – Neuroendocrine prostate cancer?) tumors, the alteration frequency of SPIN1 in this large cancer panel is very low (and includes deletions) – and is likely not above the general mutational noise seen in cancers. How does Figure 6—figure supplement 2 look when only wt p53 tumors are selected?

NEPC stands for Neuroendocrine prostate cancer. We agree with the reviewer that except NEPC, the overall frequency of SPIN1 copy number alteration is not high among other cancer types. However, as the reviewer mentioned in his or her point 1, the mRNA overexpression database is usually suggestive, but not conclusive. This is also evidenced by our Western blot analysis of human colon tissues (response to comment 1) that showed significantly higher levels of the SPIN1 protein in tumor samples than that in normal tissues (Figure 6—figure supplement 3). Further validation of SPIN1 protein expression in other cancer types would be our future study.

0In the currently available NEPC data, since the SPIN1-amplified sample size with detected mutant and wild type p53 is too small, it is difficult to make any significant conclusion. However, we will investigate this issue in the future by collaborating with Drs. David Margolin and Li Li at Oschner (as we just started to collaborate with them on this as shown in Figure 6—figure supplement 3) and by continuing our monitoring the updated cancer genomic database.